# The Association between Socioeconomic Profiles, Attitudes, and Knowledge of Dairy Farmers Regarding Somatic Cell Count and Milk Quality

**DOI:** 10.3390/ani13172787

**Published:** 2023-09-01

**Authors:** Cristina Simões Cortinhas, Carlos Eduardo Fidelis, Neelesh Sharma, Marcos Veiga dos Santos

**Affiliations:** 1Department of Animal Nutrition and Production, School of Veterinary Medicine and Animal Sciences, University of São Paulo, Pirassununga 13653-900, SP, Brazil; cristina.cortinhas@dsm.com (C.S.C.); carlosfidelis@usp.br (C.E.F.); 2Division of Veterinary Medicine, Faculty of Veterinary Sciences & Animal Husbandry, Sher-e-Kashmir University of Agricultural Sciences & Technology of Jammu, R.S. Pura, Jammu 181102, India; drneelesh_sharma@yahoo.co.in

**Keywords:** mastitis, milk quality, socioeconomic characteristics

## Abstract

**Simple Summary:**

Mastitis is a costly disease in dairy animals and significantly affects milk quality and quantity. The milk somatic cell count (SCC) is an acceptable method for diagnosing intramammary infections and evaluating the quality of milk. Mastitis is a multifactorial disease augmented by poor farm hygiene and milking practices. Therefore, the aim of the present study was to understand dairy farmers’ knowledge about mastitis control, farm hygiene, milking practices, teat dipping, milk quality, milk SCC, the total bacterial count, social responsibility, etc., and its association with the bulk tank SCC using a structured questionnaire. The socioeconomic profiles, attitudes, and knowledge of farm managers from 63 dairy farms were analyzed. It was concluded that various parameters, such as the economic impact of mastitis, milking procedures, methods for subclinical mastitis detection, the awareness of the legal requirements of the SCC, and the total bacterial count (TBC), were associated with the SCC in bulk tank milk of dairy herds.

**Abstract:**

The primary objective of this study was to investigate the association between the bulk tank somatic cell count (SCC) and dairy farmers’ knowledge about milk quality, mastitis control, and their socioeconomic characteristics. Additionally, we estimated the association between the bulk tank SCC and bulk tank milk hygienic quality in dairy herds. Bulk tank milk samples from 120 dairy herds enrolled in the milk quality payment program were collected for the determination of the SCC, the total bacterial count (TBC), the preliminary incubation count (PIC), the laboratory pasteurization count (PC), and the coliform count (CC). Based on the bulk tank SCC results, 63 herds were selected and categorized into three groups: (a) low SCC: ≤250,000 cells/mL (*n* = 16); (b) medium SCC: >250,000 ≤ 400,000 cells/mL (*n* = 24); and (c) high SCC: >400,000 cells/mL (*n* = 23). Socioeconomic profiles, attitudes, and knowledge of somatic cell count and milk quality were assessed using previously tested questionnaires, which were used to interview the herd managers of the 63 selected dairy herds. Among the findings, 87.0% of the dairy herds had an SCC < 400 × 10^3^ cells/mL, and presented milk production as the main economic activity of the farm, whereas only 47.0% of dairy herds with an SCC > 400 × 10^3^ cells/mL had milk production as the main economic activity of the farm (*p* < 0.031). In a total of 95% of the selected herds (*n* = 60), milking machines were used, with pipeline milking machines being more predominant, accounting for 70.8% in herds with a medium somatic cell count (SCC) and 78.3% in herds with a high SCC, as opposed to herds with a low SCC at 50% (*p* < 0.031). The frequency of dairy producers’ awareness of the maximum legal requirements for the bulk tank SCC and TBC was higher in herds with a higher SCC than in herds with a medium SCC. In conclusion, the results of this study indicate a significant association between the bulk tank somatic cell count (SCC) and dairy farmers’ knowledge about milk quality, mastitis control, and socioeconomic characteristics. These findings emphasize the importance of knowledge and the socioeconomic profiles of dairy farms in maintaining and enhancing milk quality in dairy herds.

## 1. Introduction

Bovine mastitis stands as the most prevalent and costly production disease in dairy herds globally [1]. The somatic cell count (SCC) is widely used as an indicator of milk quality, and at the dairy herd level, the bulk tank milk SCC can be used to estimate the prevalence of subclinical mastitis [2]. Herds producing milk with a high SCC have increased proteolytic and lipolytic enzyme activities, which reduce milk yield for cheese production and negatively affect dairy products’ flavor and shelf life [3]. In order to reduce the losses caused by poor milk quality, the dairy industry has implemented milk quality payment programs based on the SCC, which result in bonuses and penalties on milk prices, to provide incentives for dairy producers to improve milk quality [4].

Effective control practices to reduce the prevalence of subclinical mastitis and SCC depend on human factors, such as farmer motivation and behavior, as well as technical knowledge, to prevent mastitis. Previous studies evaluated how farmers’ attitudes and knowledge interfere with the bulk tank SCC and the incidence of mastitis. For example, Jansen et al. [5] studied 336 dairy herds in the Netherlands, and they reported that both the behavior and attitudes of farmers explained 48% of the variance in the bulk tank SCC, 31% of the incidence of clinical mastitis, and 23% of the incidence of subclinical mastitis. A recent study reported that Ontario dairy producers demonstrated a high level of concern about udder health, exhibited positive attitudes toward udder health management practices, and emphasized the importance of cow comfort and herd welfare [6].

Although previous studies evaluated the attitudes and knowledge of the SCC and the incidence of mastitis in dairy herds with or without the implementation of mastitis control programs, few studies evaluated the technical knowledge and opinion of dairy farmers about the causes of mastitis, losses, and control practices utilized for mastitis management on farms that submitted milk samples to milk quality payment programs. This type of evaluation could be used to detect differences not only in knowledge but also in the awareness and perception of dairy farmers regarding udder health improvement [7].

However, despite the growing body of knowledge on the impact of the socioeconomic factors, knowledge, and attitudes of dairy farmers on the SCC and milk quality, limited research has examined these relationships in a comprehensive manner. The objective of this study was to estimate the association between the bulk tank SCC and dairy farmers’ knowledge about milk quality and mastitis control, and their socioeconomic characteristics. Furthermore, this study also aimed to estimate the correlation between the bulk tank SCC and other hygienic indicators of bulk tank milk quality in dairy herds.

## 2. Materials and Methods

### 2.1. Selection of Dairy Herds

In the present study, 120 dairy farms were selected from a pool of 498 dairy farm members of a dairy industry in the southern region of Minas Gerais State, Brazil. This dairy industry has implemented a milk quality payment program that is based on the following criteria: the total bacterial count (TBC), SCC, and protein content. Briefly, the payment program utilizes the geometric mean results for the TBC, SCC, and protein content derived from five bulk tank milk samples per month, which were used to calculate either a premium or penalty on the milk price paid to the dairy producer.

### 2.2. Milk Sample Collection

Bulk tank milk samples from 120 herds were collected on a weekly basis over a two-month period (August and September 2011) to perform microbiological tests (total bacterial count—TBC; preliminary incubation count—PIC; pasteurized count—PC; and coliform count—CC) and composition analyses (fat, protein, and total solids). A total of 816 milk samples were collected by trained milk haulers, who ensured the proper collection of bulk tank milk samples after 5 to 10 min of agitation of milk, followed by the refrigeration of the milk samples at 4.0 °C immediately after milk collection [8]. For milk evaluation, samples were simultaneously collected, employing distinct procedures for each analysis. For the microbiological analysis, the milk samples were collected in sterile vials without preservatives, frozen at −20 ° C, and stored until analysis (maximum of six days). For the milk composition analysis, milk samples were collected in vials containing 8 mg of the preservative bronopol (2-bromo-2-nitro-1,3-propanediol), which were then stored at 4 °C for a maximum of five days after collection.

### 2.3. Somatic Cell Count and Microbiological Analysis

Somatic cell counts were analyzed using the fluoro-opto-electronic counter method in (Somacount 300^®^, Bentley Instruments Inc., Chasca, MN, USA. The analysis was carried out in a certified laboratory participating in the Brazilian Milk Quality Network (BMQN), with results expressed as one thousand somatic cells per mL. For microbiological analysis, the milk samples were diluted (1:10 in 1% buffered peptone water) and seeded (50 µL of diluted milk) in standard plate agar (Plate Count Agar, Merck, Darmstadt, Germany), using Easy Spiral Pro (Interscience, St. Nom, France). The colony count was performed using an automatic colony counter SCAN 500 (Interscience, St. Nom, France) after incubating at 32 °C (±1 °C) for 48 h. For performing the preliminary incubation count, 3 mL of milk samples were incubated for 18 h at 12.8 °C, and after this procedure, the samples were maintained at 4 °C, diluted (1:10), plated in standard plate count agar, and incubated for 48 h at 32 °C. Colony counting was carried out using an automatic colony counter SCAN 500 [9].

The pasteurized count was determined following the laboratory pasteurization of 3 mL of milk samples for 30 min at 62.8 °C in a water bath. After this step, milk samples were submitted to a thermal shock in ice and water, diluted (1:10), plated on standard count agar, and then incubated for 48 h at 32 °C. Colony counting was carried out using an automatic colony counter SCAN 500 [9].

The milk samples were diluted (1:10), and 1 mL of the diluted sample was inoculated onto Petrifilm plates to determine the coliform count (3M, St. Paul, MN, USA) [10]. Petrifilm plates were incubated for 24 h (±2 h) at 32 °C (±1 °C), and colony counting was carried out manually.

### 2.4. Data Collection on Dairy Farms

A total of 57 herds were excluded from this study due to contaminated samples (*n* = 15), and an additional 42 herds discontinued delivering milk to the dairy industry selected for this study. Thus, 63 farms were visited to conduct on-farm questionnaires about the socioeconomic profile, attitudes, and knowledge of dairy farmers. These questionnaires were designed based on previous studies [5,11,12]. The questions were tailored for the region where the study was conducted and were previously tested in an interview on one of the dairy farms.

The questionnaire was consistently administered by the same researcher, and the questions were required to be answered by the herd managers, and included the following topics: (a) socioeconomic information (aimed to stratify farmers according to their income, volume of milk production, labor force, and technical assistance); (b) the measures used for the prevention and treatment of mastitis; and (c) knowledge about SCC, mastitis, TBC and milk quality legislation.

The selected dairy herds (*n* = 63) were categorized into three groups based on the average of SCC results from eight samples obtained during consecutive weeks over two months: (a) low SCC: ≤250,000 cells/mL (*n* = 16); (b) medium SCC: >250,000 ≤ 400,000 cells/mL (*n* = 24); and (c) high SCC: >400,000 cells/mL (*n* = 23). The threshold of SCC 250,000 cells/mL was used to classify herds with low SCC [9], and those exceeding 400,000 cells/mL were categorized as herds with high SCC [13].

## 3. Statistical Analyses

Prior to statistical analysis of the results, data were tested for residual normality using the Shapiro–Wilk test using SAS version 9.1 (SAS Institute Inc., Cary, NC, USA), (PROC UNIVARIATE) and the variances were compared using the F test. The analysis of the response frequencies was performed using the Cochran–Mantel–Haenszel Test. Data obtained from the microbiological testing (TBC, PIC, PC, and CC) and SCC were converted to a logarithmic scale and were submitted to the variance analysis (PROC ANOVA) using the following model:*Y_i_* = *μ* + ccs*_i_*+ ε*_ij_*
where

*Y_i_* = variable response (TBC, PIC, PC, and CC);

*μ* = overall average;

ccs*_i_* = effect the *i*-th SCC group;

ε*_ij_* = random error term.

The comparison of mean was performed by the Scheffé test, allowing for the contrast between means with different numbers of observations per treatment. Pearson correlation coefficients were calculated between milk quality parameters and the SCC (PROC CORR). The results of SCC and TBC were expressed in a logarithmic scale, and *p*-values < 0.05 were considered significant.

## 4. Results

In this study, dairy farms received bonuses or penalties based on their bulk tank milk quality results. Specifically, farms were paid $0.010/kg if the bulk milk tank presented TBC in the range of 1–50 × 10^3^ CFU/mL. On the other hand, if the bulk milk tank TBC was higher than 1000 × 10^3^ CFU/mL, the dairy farms received a penalty of $0.010/kg of milk. Similar to TBC, the SCC and milk protein content were used to incentivize or penalize the price of milk paid to dairy farms, with values ranging from $0.003 to $0.010/kg (Table 1).

Regarding the socioeconomic characteristics of dairy farmers, the bulk tank SCC had a significant effect (*p* < 0.031) on the frequency of answers about the primary economic activity of the farm and the type of milking equipment used (Table 2). Specifically, for nearly 87% percent of the herds with low and medium SCC, farmers reported they had milk production as the primary economic activity of the farm. In contrast, for only 47% of herds with high SCC, milk production was reported as the primary economic activity. Additionally, farmers with low and medium SCC indicated a tendency (*p* = 0.067) to have milk production as the primary source of income for their families.

Mechanical milking was used on 95% of the herds, while only 5% used manual milking. Pipeline milking was the type of milking equipment in 78.3% of herds with high SCC, 70.8% of herds with medium SCC, and only 50% of herds with low SCC. No significant differences were observed in terms of the frequency of the responses across different variables, including the education level of the farmer or the individual responsible for the herd management, average monthly income, the daily milk production of each herd, the type of labor force involved in milk production (family, permanent hire, and temporary hire) and the type of technical assistance (veterinary, cooperatives, or other) used among the three groups of herds.

Regarding the management practices for mastitis prevention and control (Table 3), there was a significant difference (*p* < 0.042) in the frequency of using the California Mastitis Test (CMT) to identify subclinical mastitis among the evaluated groups. CMT is a cow-side test that can be used to estimate the SCC at the quarter level. Due to its ease of application, this method can be conveniently used before milking cows. The frequency of using the CMT was 68.7% in herds with low SCC, 45.8% in herds with medium, and only 34.8% in herds with high SCC. Moreover, herds with high SCC exhibited a tendency (*p* = 0.079) to use antibiotics more frequently for treating clinical mastitis than herds with medium and low SCC. A similar tendency (*p* = 0.057) was observed in the differences in using mastitis treatment protocols among herd groups based on the bulk tank SCC. The majority of herds with low SCC (66.7%) and medium (73.9%) SCC used intramammary antimicrobials with systemic antimicrobials.

The effect of bulk tank SCC was not significant among herds for the following variables: the recording of clinical mastitis cases or protocols, the use of anti-inflammatory drugs for the treatment of clinical mastitis, dry cow therapy, the use of SCC to diagnose individual cases of clinical mastitis, the use of microbiology culture to identify the major pathogens, the culling of cows due to chronic mastitis, vaccinations, and the segregation of cows with mastitis.

In this study, there was no effect of the bulk tank SCC on the technical knowledge of dairy managers about mastitis (Table 4). Only 8 (12.7%) dairy managers answered that they had no knowledge about mastitis, whereas 48 dairy managers (76.2%) correctly defined mastitis as an infection or the inflammation of the mammary gland, and 4 dairy managers (6.3%) provided other responses. All dairy managers acknowledged that cows affected by mastitis experience pain. Regarding subclinical mastitis, the majority of the dairy managers (77.8%) correctly responded that it does not cause visible changes in milk, and 65% of dairy managers correctly answered that the SCC should be used as an indicator of subclinical mastitis. Regarding the identification of the primary risk factor for mastitis, 47 dairy managers (74.6%) responded that poor hygiene was the leading risk factor, 2 (3.2%) associated mastitis with reduced feed intake, 6 (9.5%) associated mastitis with other issues, and 8 (12.6%) were unsure.

The results of the assessment of farmers’ technical knowledge about bacterial contamination, milk quality, and Brazilian legislation are presented in (Table 5). There was a significant difference (*p* < 0.013) in the frequency of affirmative responses about the legal requirements of SCC and TBC in herds with high SCC (56.5%), as compared to herds with medium (29.2%) or low (18.7%) SCC. The frequency of responses to the questions about the causes of milk bacterial contamination, as well as questions regarding who considers milk quality important, was similar among the respondents who participated in courses about mastitis and milk quality. A majority of dairy managers (85.7%) correctly defined the bacterial contamination of milk as the total amount of bacteria present in milk, while eight (12.7%) were unsure. Regarding the causes of bacterial contamination, 44 participants (77.3%) associated it with general hygienic conditions, 3 participants (5%) associated it with the presence of sick animals, 7 (8.3%) associated it with the use of manual milking, 2 (3.3%) associated it with both general hygiene and sick animals, and 6 (10%) were uncertain. When asked about the stakeholders most interested in milk quality, 2 (3.4%) indicated dairy farmers, 3 (5.1%) designated the dairy industry, and 11 (18.7%) responded the consumers. Furthermore, 21 participants (35.6%) considered dairy farmers, consumers, and the dairy industry to be most interested in milk quality, while 22 (37.3%) attributed this interest to dairy farmers, consumers, the dairy industry, and the government. Additionally, 25 (39.7%) respondents had participated in training about mastitis and milk quality.

The average milk quality indicators (TBC, PIC, PC, and CC) of herds with low, medium, and high SCC, evaluated over a two-month period, are shown in Table 6. There was a significant difference (*p* < 0.001) in all the hygiene indicators analyzed (TBC, PIC, PC, and CC) among herds with high, medium, and low SCC. Herds with high SCC exhibited higher values for all indicators compared with herds with medium and low SCC. Similarly, herds with medium SCC had higher values for all indicators than herds with low SCC.

The coefficient of correlations between the bulk tank SCC and the milk hygienic quality were as follows: TBC (r = 0.310; *p* < 0.001), PC (r = 0.305; *p* < 0.001), CC (r = 0.232; *p* < 0.001), and PIC (r = 0.215; *p* < 0.001).

## 5. Discussion

Mastitis is a multifactorial disease leading to increased SCC, which is a key factor in determining not only the milk quality for milk payment programs by dairy industries but also in adhering to the legal limits of human milk consumption. However, solely relying on the quantitative analysis of primary management practices only partially explains the variation in the prevalence of this disease among dairy herds [6]. In this study, we identified a significant association between the bulk tank SCC and several factors, including the importance of milk as the main economic activity of dairy farms, the type of milking equipment used in dairy herds, the awareness of the legal limits of SCC and TBC, and the frequency of using CMT for mastitis detection.

In this study, in the majority of dairy farms with herds with low (86.7%) and medium SCC (87.5%), milk production was the primary economic activity. Conversely, only 47.6% of dairy farms with herds with high SCC had milk production as the main economic activity. Our results indicate that the majority of low- and medium-SCC dairy farms dedicated most of their efforts to milk production instead of other activities, potentially leading to a higher level of concern regarding the management practices aiming to reduce SCC, particularly because these farms received an additional premium payment for low bulk tank SCC. Previously, studies demonstrated a significant association between the adoption of milk quality payment programs and the reduction in both the SCC and TBC of bulk tank milk [4]. Moreover, the adoption of such programs motivates dairy farmers to implement management practices for mastitis control. However, the motivation to improve mastitis management may differ among individual farmers [14].

The results of this study revealed that, for 95% of the selected herds, mechanical milking was used, while in only a small proportion (5%) of herds, manual milking was used. The inadequate use of the milking equipment may be a potential risk factor for new intramammary infections, especially when the equipment has maintenance deficiencies [15]. Interestingly, Barbosa et al. [16] described that herds in Minas Gerais, Brazil, for which manual milking was used had lower SCC than herds subjected to mechanical milking [16].

The use of pre-milking management practices with the aim of reducing the SCC, teat contamination, and the adequate stimulation of milk ejection has been a common approach. On the other hand, the use of suckling calves before milking to stimulate milk ejection is a common practice in herds of crossbred milking cows (Holstein X Zebu). According to Barbosa et al. [16], the presence of suckling calves may reduce the residual milk after milking, which diminishes the incidence of mastitis. Conversely, Araújo et al. [17] did not find differences in SCC among Gir cows stimulated via suckling calves and cows stimulated via the exogenous administration of oxytocin. In the present study, in 11% of all herds with mechanical milking systems, suckling calves were used before milking to stimulate milk ejection, and among these herds, 9.5% had low and medium SCC, while 1.5% of herds had high SCC [17].

The CMT (California Mastitis Test) is a method to indirectly estimate the SCC at the quarter level [18]. It is a simpler alternative mastitis diagnosis method compared with electronic cell counting. In our study, the frequency of CMT usage was higher among farmers with low SCC than among those with medium and high SCC. This finding suggests that farmers with lower SCC values are more proactive in detecting subclinical mastitis cases and taking appropriate measures. However, no differences were observed in the frequency of using individual electronic SCC tests among the three evaluated SCC groups. These findings differ from the results reported by Oliveira et al. [19], who found that only 2% of farms used the CMT, and no correlation was observed between the bulk tank SCC and the frequency of CMT use among 50 dairy farms in Paraiba, Brazil [19].

Both the CMT and SCC can be used to detect subclinical mastitis; however, the CMT has several advantages, including the ease of use in the pre-milking process, its cost-effectiveness, and the availability of immediate results. The CMT allows for the rapid detection of subclinical mastitis, which is crucial in making informed decisions about mastitis control. On the other hand, the SCC is a more expensive alternative, and additionally, it requires sending the samples to a specialized laboratory, resulting in longer turnaround times. Regardless of the subclinical mastitis diagnosis method, the results of this study indicate that herds using such diagnostic methods had lower bulk tank milk SCC. Nevertheless, no difference was observed among the SCC groups concerning other mastitis control measures, such as the use of microbiological culture or the culling of cows with mastitis.

In this study, herds with high SCC had a tendency (*p* = 0.079) of higher frequency of antibiotic use for treating clinical mastitis. Effective mastitis control programs prioritize prevention over treatment, with the therapy being used as part of a mastitis control program for clinical cases. Clinical mastitis treatment aims to return cows to milk production and to eliminate the mastitis-causing pathogen in the infected quarter [20]. In Brazil, Souza et al. [21] observed that in the majority (97.7%) of herds evaluated in Zona da Mata Mineira, the treatment of clinical cases of mastitis was the primary procedure of mastitis control. A recent study by Tomazi et al. [22] reported that 97% of clinical mastitis cases were treated with antimicrobials, although a wide variation in the frequency of antimicrobial use was observed among Brazilian dairy herds.

In our study, treating clinical mastitis with intramammary antibiotics was more frequent (*p* = 0.057) in herds with higher SCC, and the association of intramammary and systemic antibiotic therapy was higher in herds with lower SCC. This suggests that the use of antimicrobials in clinical mastitis treatment is influenced by the herd’s SCC level, potentially due to a higher number of clinical cases in herds with high SCC.

In this study, the awareness about the legal limits for SCC and TBC of bulk tank milk was more common among farm managers of herds with high SCC than among those herds with medium and low SCC. These results could be attributed to the penalties in terms of milk price for herds exceeding the SCC legal limit. However, it should be considered that knowledge about legal requirements does not ensure the implementation of management practices to reduce the SCC. In addition, the low penalty received by the SCC > 600,000 cells/mL (SCC between 600,000 and 750,000 cells/mL–$0.003/L and above 750,001 cells/mL–$0.007/L) may not be sufficient to encourage dairy farmers to implement measures to reduce the SCC.

In this study, there was no association between the bulk tank SCC and the level of knowledge of dairy farmers about mastitis (clinical, subclinical, and SCC definitions, as well as causes of mastitis). Likewise, the level of technical knowledge of dairy managers about bacterial contamination (definition, causes, and importance) was not associated with the bulk tank of SCC. However, farm personnel with high-SCC herds had greater awareness of the legal requirements of SCC and TBC. The potential penalties imposed on high-SCC farms may motivate farmers to better understand the reasons behind their penalties and ensure compliance with the regulations.

Our study revealed that herds with high SCC presented higher total bacteria count, PIC, PC, and CC. A low positive correlation was observed between the SCC and the hygienic milk quality indicators, namely, TBC, PIC, PC, and CC. These results were similar to previous studies such as Rysanek et al. [23], who observed a low correlation between SCC, TBC, and bulk tank milk’s CC. Jayarao et al. [13] also reported a low correlation between SCC and other microbiological analyses (PIC, PC, CC, environmental streptococci, and coagulase-negative Staphylococci count) of raw milk. These results suggest that relying solely on the bulk tank milk SCC is insufficient for accurately assessing milk hygiene, since other factors such as milking routine, the cooling efficiency during milk storage, and transportation equipment can impact the overall hygienic condition of milk [24].

## 6. Conclusions

In conclusion, our results highlight a significant association between the bulk tank somatic cell count (SCC) and the knowledge of dairy farmers about milk quality, mastitis control, and their socioeconomic characteristics. Factors such as the importance of milk as an economic activity on the farm, milking equipment type, the use of the CMT for subclinical mastitis detection, and the awareness of dairy farmers about the SCC and TBC legal limits, were found to be associated with the bulk tank SCC. Additionally, there was a positive correlation between the hygienic quality of milk and the bulk tank SCC in dairy herds. These findings emphasize the pivotal role of knowledge and the socioeconomic profile of dairy farms in producing high-quality milk.

## Figures and Tables

**Table 1 animals-13-02787-t001:** Premium or penalty (USD/kg of milk) according to the monthly average of bulk tank milk total bacteria count (TBC) and somatic cell count.

Payment (USD/kg)	TBC (×10^3^ CFU/mL)	SCC (×10^3^ cells/mL)
0.010	1–50	-
0.007	>50–100	1–200
0.003	>100–200	>200–400
0.000	>200–500	>400–600
−0.003	>500–750	>600–750
−0.007	>750–1000	>750
−0.010	>1000	-
**Payment (USD/kg)**	**Protein (%)**	
0.004	3.17–3.19	-
0.002	3.14–3.16	-
0.001	3.11–3.13	-
0.000	3.01–3.10	-
−0.003	2.95–3.00	-
−0.008	2.90–2.94	-
−0.016	2.85–2.89	-

**Table 2 animals-13-02787-t002:** Socioeconomic characteristics according to the monthly average of bulk tank milk SCC (*n* = 63 farms).

Variable	Answer	Bulk Tank Milk SCC (10^3^ cells/mL)			*p*
		Low *n* (%)	Medium *n* (%)	High *n* (%)	
Education level (total years of formal education)	Do not have	0 (0.0)	0 (0.0)	1 (4.35)	0.673
	Up to 5 years	6 (37.5)	10 (41.7)	7 (30.4)	
	Up to 8 years	4 (25.0)	5 (20.8)	7 (30.4)	
	Up to11 years	2 (12.5)	5 (20.8)	5 (21.7)	
	College level	4 (25.0)	4 (16.7)	3 (13.0)	
Average monthly income (USD)	<456	0 (0.0)	0 (0.0)	0 (0.0)	0.815
	From 266 to 531	2 (12.5)	3 (13.0)	2 (8.7)	
	From 532 to 797	5 (31.2)	11 (47.8)	8 (34.8)	
	From 798 to 1131	1 (6.3)	4 (17.4)	2 (8.7)	
	>1132	8 (50.0)	5 (21.7)	11 (47.8)	
Main economic activity of the farm ^1^	Agriculture	1 (6.7)	3 (12.5)	10 (47.6)	0.031
	Milk production	13 (86.7)	21 (87.5)	10 (47.6)	
	Beef cattle production	1 (6.7)	0 (0.0)	0 (0.0)	
	Others	0 (0.0)	0 (0.0)	1 (4.8)	
Economic importance of milk production to the family income	Main activity	13 (81.2)	20 (83.3)	11 (47.8)	0.067
	Second activity	1 (6.25)	2 (8.3)	6 (26.1)	
	Complementary	1 (6.3)	1 (4.2)	4 (17.4)	
	Unimportant	1 (6.3)	1 (4.2)	2 (8.7)	
Average milk production (L milk/day/farm)	101 ≥ 500	8 (50.0)	6 (25.0)	9 (39.1)	0.487
	501 ≥ 1000	2 (12.5)	9 (37.5)	10 (43.5)	
	1001 ≥ 2000	1 (6.2)	5 (20.8)	1 (4.3)	
	2001 ≥ 5000	5 (31.2)	1 (4.2)	2 (8.7)	
	>5000	0 (0.0)	3 (12.5)	1 (4.3)	
Type of labor (involved with milk production)	Family labor	5 (31.2)	5 (20.8)	3 (13.1)	0.750
	Permanent labor	7 (43.7)	15 (62.5)	17 (73.9)	
	Permanent and temporary labor	4 (25.0)	4 (16.7)	3 (13.0)	
Type of milking	Manual	1 (6.2)	2 (8.3)	0 (0.0)	0.031
	Bucket with suckling calves during milking	4 (25.0)	2 (8.3)	1 (4.3)	
	Bucket without calves during milking	3 (18.7)	3 (12.5)	4 (4 (17.4)	
	Pipeline	8 (50.0)	17 (70.8)	18 (78.3)	
Type of technical assistance	Private vet	8 (50.0)	12 (50.0)	12 (52.2)	0.835
	Cooperative	3 (18.7)	0 (0.0)	2 (8.7)	
	Other	5 (31.3)	10 (41.7)	8 (34.8)	
	Do not have	0 (0.0)	2 (8.3)	1 (4.3)	

^1^ Three farmers with 50% of agriculture and 50% of milk production as the main activity were excluded.

**Table 3 animals-13-02787-t003:** Frequency of management practices used for mastitis prevention and control in herds with low, medium, and high SCC.

Variable	Response	Bulk Tank Milk SCC (10^3^ cells/mL)			*p*
		<250	250 ≤ 400	>400	
		*n* (%)	*n* (%)	*n* (%)	
Adequate records of clinical mastitis cases	Yes	10 (62.5)	15 (62.5)	10 (43.5)	0.209
	No	6 (37.5)	9 (37.5)	13 (56.5)	
Adequate records of protocols for mastitis treatment	Yes	10 (62.5)	17 (20.8)	12 (52.2)	0.441
	No	6 (37.5)	7 (29.2)	11 (47.8)	
Use of antibiotics for clinical mastitis treatment	Yes	14 (85.5)	23 (95.8)	23 (100.0)	0.079
	No	1 (12.5)	1 (4.2)	0 (0.0)	
Type of mastitis treatment protocol	Intramammary antimicrobial1 to 3 days	3 (20.0)	3 (13.1)	11 (50.0)	0.057
	Intramammary antimicrobial IMM until healing	2 (13.3)	1 (4.3)	1 (4.5)	
	Intramammary + systemic antimicrobial	10 (66.7)	17 (73.9)	10 (43.5)	
	Only systemic antimicrobial	0 (0.0)	2 (8.7)	0 (0.0)	
Use of anti-inflammatory drugs for mastitis treatment	No	4 (25.0)	6 (25.0)	7 (30.4)	0.847
	Only severe mastitis cases	11 (68.7)	13 (54.1)	12 (52.17)	
	All mastitis cases	1 (6.3)	5 (20.8)	4 (17.4)	
Use of dry cow treatment	Yes	14 (87.5)	21 (87.5)	18 (78.3)	0.401
	No	2 (12.5)	3 (12.5)	5 (21.7)	
Use of California Mastitis Test	Yes	11 (68.7)	11 (45.8)	8 (34.8)	0.042
	No	5 (31.3)	13 (54.2)	15 (65.2)	
Subclinical mastitis diagnosed by individual SCC	Yes	4 (25.0)	11 (45.8)	5 (21.7)	0.673
	No	12 (75.0)	13 (54.2)	18 (78.3)	
Use of microbiological culture to identify mastitis causative agents	Yes	4 (25.0)	7 (29.2)	6 (26.1)	0.968
	No	12 (75)	17 (70.8)	17 (73.9)	
Culling cows by mastitis cases	Yes	11 (68.7)	22 (91.7)	15 (65.2)	0.615
	No	5 (31.3)	2 (8.3)	8 (34.8)	
Use of vaccine for mastitis	Yes	5 (31.3)	4 (17.4)	4 (17.4)	0.331
	No	11 (68.7)	19 (82.6)	19 (82.6)	
Grouping cows in lots by clinical and subclinical mastitis	Yes	4 (25.0)	9 (37.5)	5 (21.7)	0.722
	No	12 (75.0)	15 (62.5)	18 (78.3)	

**Table 4 animals-13-02787-t004:** Frequency of responses of technical knowledge about mastitis in herds with low, medium, and high SCC.

Question	Response	Bulk Tank Milk SCC (10^3^ cells/mL)			*p*
		<250 *n* (%)	250 ≤ 400 *n* (%)	>400 *n* (%)	
Mastitis definition	Mammary gland infection or inflammation	13 (81.3)	16 (66.7)	19 (82.6)	0.924
	Disease of the udder	1 (6.3)	2 (8.3)	0 (0.0)	
	Do not know	1 (6.3)	5 (20.8)	2 (8.7)	
	Other answers	1 (6.3)	1 (4.2)	2 (8.7)	
Cows with mastitis feel pain	Yes	16 (100.0)	24 (100.0)	23 (100.0)	-
	No	0 (0.0)	0 (0.0)	0 (0.0)	
Subclinical mastitis definition	Mastitis does not cause visible changes in milk	13 (81.3)	20 (83.3)	16 (69.6)	0.429
	Other answers	0 (0.0)	0 (0.0)	1 (4.3)	
	I do not know	3 (18.7)	4 (16.7)	6 (26.1)	
Somatic Cell Count definition	Indicative of subclinical mastitis	10 (62.5)	18 (75.0)	13 (56.5)	0.781
	Other answers	1 (6.3)	0 (0.0)	3 (13.1)	
	Do not know	5 (31.3)	6 (25.0)	7 (30.4)	
Causes of mastitis	Lower feed intake	0 (0.0)	1 (4.2)	1 (4.3)	0.412
	General hygiene	10 (62.5)	20 (83.3)	17 (73.9)	
	Milking equipment hygiene	1 (6.3)	1 (4.2)	1 (4.3)	
	Do not know	3 (18.7)	1 (4.2)	1 (4.4)	
	Others	2 (12.5)	1 (4.2)	3 (13.1)	

**Table 5 animals-13-02787-t005:** Frequency of responses of technical knowledge about bacterial contamination and legislation in herds with low, medium, and high SCC.

Question	Response	Bulk Tank Milk SCC (10^3^ cells/mL)			*p*
		<250 *n* (%)	250 ≤ 400 *n* (%)	>400 *n* (%)	
Definition of “bacterial contamination”	Number of bacteria in milk	14 (87.5)	20 (83.3)	20 (87.0)	0.791
	Other answers	1 (6.3)	0 (0.0)	0 (0.0)	
	Do not know	1 (6.3)	4 (16.7)	3 (13.0)	
Causes of bacterial contamination ^1^	General Hygiene	13 (81.3)	16 (69.6)	15 (71.4)	0.674
	Sick Animals	0 (0.0)	2 (8.7)	1 (4.8)	
	Manual milking	2 (12.5)	1 (4.3)	2 (9.5)	
	General hygiene and sick animals	0 (0.0)	0 (0.0)	2 (9.5)	
	Do not know	1 (6.3)	4 (17.4)	1 (4.8)	
For whom the quality of milk is more important? ^2^	Farmers	1 (6.3)	0 (0.0)	1 (4.6)	0.863
	Dairy	1 (6.3)	0 (0.0)	2 (9.1)	
	Consumers	4 (25.0)	3 (14.3)	4 (18.2)	
	Farmers, dairy consumers	5 (31.3)	9 (42.9)	7 (31.8)	
	Farmers, dairy consumers, and the government	5 (31.3)	9 (42.9)	8 (36.4)	
Participation in any education/training on mastitis and milk quality?	Yes	5 (31.2)	10 (41.7)	10 (43.5)	0.466
	No	11 (68.3)	14 (58.3)	13 (56.5)	
Knowledge about the legal requirements for somatic cell count and total bacterial count of milk	Yes	3 (18.7)	7 (29.2)	13 (56.5)	0.013
	No	13 (81.3)	17 (70.8)	10 (43.5)	

^1^ Three producers with multiple answers were excluded from the analysis; ^2^ four producers with multiple answers were excluded from the analysis.

**Table 6 animals-13-02787-t006:** Total bacterial count (TBC), preliminary incubation count (PIC), pasteurized count (PC), total coliform count (CC), and somatic cell count (SCC), grouped by somatic cell count.

Somatic Cell Count					
Variable	≤250	>250 ≤ 400	>400	SEM ^1^	*p*
TBC, UFC/mL (log)	32,708.8 ^c^ (4.51)	86,744.0 ^b^ (4.93)	103,088.7 ^a^ (5.01)	0.035	<0.001
PIC, UFC/mL (log)	192,829.5 ^c^ (5.28)	354,598.3 ^b^ (5.54)	444,990.2 ^a^ (5.64)	0.041	<0.001
PC, UFC/mL (log)	857.7 ^c^ (2.93)	1174.6 ^b^ (3.06)	2939.4 ^a^ (3.46)	0.004	<0.001
CC, UFC/mL (log)	74.21 ^c^ (1.87)	258.4 ^b^ (2.41)	1678.2 ^a^ (3.22)	0.050	<0.001

^1^ Standard error of the mean. Means followed by the same letter on the same line do not differ (*p*> 0.05) based on the Scheffé test.

## Data Availability

The data presented in this study are available on request from the corresponding author.

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
