# Peer review of "The Association between Socioeconomic Profiles, Attitudes, and Knowledge of Dairy Farmers Regarding Somatic Cell Count and Milk Quality"

_animals, 2023, doi:10.3390/ani13172787_

Round 1
Reviewer 1 Report
Dear Author,
Thank you for sending in this manuscript. There were numerous, but minor, grammatical errors made. I have only made some corrections, as shown in the attachment. Overall, an interesting read with good flow despite the errors.

See comments in the Editors section below
Author Response
Dear Editor-in-Chief and reviewers,
I am writing to submit the revised version of our manuscript titled "Association Between Socioeconomic Profile, Attitudes and Knowledge of Dairy Farmers About Somatic Cell Count and Milk Quality" for further consideration in Animals Journal.
We are confident that the revised version of our manuscript now meets the high standards set by Animals Journal. We sincerely hope that you will find our research suitable for publication in your esteemed journal.
RESPONSE LETTER
Line 12: Please amend to: “Mastitis is a costly disease of dairy animals and significantly affects milk quality and quantity.”
AU: Done as suggested. See line 12.
Line 14/15: Please rephrase to: Mastitis is a multifactorial disease which is augmented by poor farm hygiene and milking practices.”
AU: Done as suggested. See lines 15 to 16.
Line 15: Please rephrase to: “Therefore, the aim of the present study was to understand the….”
AU: Done as suggested. See line 17.
Line 17: Please amend to: “…. through a structured questionnaire..”
AU: Done as suggested. See line 20.
Line 19: Please amend to: “…. 63 dairy farms were…”
AU: Done as suggested. See line 21.
Line 21: As this is the first time that the abbreviation TBC is used, give the full name followed by the abbreviation in
AU: We described the full name followed by the abbreviation. See line 23.
Line 20/21: Please rephrase: “.. awareness about legal requirements of SCC and TBC were associated with SCC in bulk tank milk of dairy herds.”
AU: Done as suggested. See line 23.
Line 30: what is meant by the words “attitudes and knowledge”. Attitudes to what and knowledge about what?
AU: We modified the sentence to better clarify. See line 34.
Line 31: Leave out “a” in the following: using a previously tested questionnaires
AU: Done as suggested. See line 34.
Line 35: Rephrase as follows: “A total of 95% of the selected herds (n = 60) used milking machines.“
AU: We modified the sentence to better clarify. See line 39.
Line 36/37: Rephrase as difficult to understand: “.. pipeline milking machine being more prevalent in 70.8% in herds with 36 medium SCC and by 78.3 % and high SCC, compared to herds with low SCC (50%; P<0.031).
AU: We modified the sentence to better clarify. See line 40.
Line 39/40: Please amend to: “In conclusion, the economic impact on milk quality and yield as a result of mastitis, …”
AU: Done as suggested. See lines 45 and 46.
Line 84: Please amend to: “498 dairy farm members…”
AU: Done as suggested. See line 90.
Results section: Mention table 1 here
AU: We added a sentence that describe table 1 here. See line 189 to 193.
Line 105-109 Was the same sample used to carry out both procedures or was each sample split at time of collection? Please mention this in your materials and methods section
AU: The samples were collected at the same time with different procedures for each analysis. We added a sentence to better clarify. See lines 114 and 115.
Line 206: Please amend “de major..” to “the major..”
AU: Done as suggested. See line 226.
Line 222: Please amend “SCC servers as..” to “SCC serves as…”
AU: Done as suggested. See line 266.
Line 256: Should the abbreviations “CP” and “CIP” be “PC” and “PIC”? Please check for these abbreviations elsewhere in the text.
AU: We revised all the abbreviations on the manuscript.
Line 327: amend to “using such diagnostic methods”
AU: Done as suggested. See line 376.
Line 328/329: “Nevertheless, no difference were observed among the three levels of SCC evaluated concerning other preventive measures of this disease.” Mention these other preventative methods for mastitis.
AU: We added a sentence that mention these other preventative methods. See line 379.
Line 347: What is “CBT” the abbreviation for? Should this be TBC?
AU: We revised all the abbreviations on the manuscript.
Reviewer-2 Comments
In this paper entitled “Association Between Socioeconomic Profile, Attitudes and Knowledge of Dairy Farmers About Somatic Cell Count and Milk Quality”, the Authors, investigate the association between the bulk tank somatic cells count and the knowledge of dairy farmers about milk quality, mastitis control, and their socioeconomic characteristics, by administering specific questionnaires to farmers. Furthermore, this study estimated the association between bulk tank somatic cells count and the hygienic quality of bulk milk on dairy farms. The socio-economic profile, attitudes and knowledge of the managers of 63 dairy farms, were analysed. The Authors conclude that the economic impact of milk of mastitis, milking procedures, methods for subclinical mastitis detection, and awareness about legal requirements of somatic cell count and total bacterial count were associated with somatic cell count in bulk tank milk of dairy herds.
This paper, in line with the subject matter of the journal, is well articulated in its treatment of the topics, with interest for the scientific community, it needs some additions that I report below line-by-line:
Line 93-95: table 1. Check the measurement intervals in table 1, e.g. 1-50, 50-100, 100-200
AU: The range values presents in table 1 were corrected, as sugested.
Line 99-100: (August and 99 September 2011). Is this date correct?
AU: This date is correct “August and September 2011”. In the comment of the reviewer “August and 99 September 2011”, the “99” refers to the line number.
Line 113-114: "Somatic cell counts were analysed by electronic counting with flow cytometry (Fossomatic FC, Foss Electric A/S Hillerod, Denmark). Add further details, e.g. internal standards, measuring range, ….
AU: We have added a sentence with a better description of the parameters used in flow cytometry. See lines 124 to 127.
Line 169: Check “eij =randon error”.
AU: Done as suggested. See line 182.
Line 174: The results are expressed as ...; specify.
AU: Done as suggested. See lines 186 and 187.
Line 194: I suggest the Authors include a short comment on the use of the CMT in the milking parlour (see also line 309).
AU: Done as suggested. See lines 215 and 216.
Line 209: Check “Monthly average income (R$)”.
AU: We modify this sentence. See in the table 2.
Line 217: Check “managers (76,2%).”
AU: Done as suggested. See line 261.
Line 219: Check “managers (6,3%).”
AU: Done as suggested. See line 263.
Line 225-226: “managers (3,2%) associated mastitis with reduced feed intake, 6 (9,5%) associated mastitis 225 with other issues, and 5 (7,9%) were unsure”. Check the %.
AU: We checked the percentage. See lines 267 to 271.
Line 263-265: Table 6. Total bacterial count (TBC), preliminary incubation count (PIC), pasteurized count (PC), total coliform count (CC) and somatic cells count (SCC), grouped by somatic cells count. Are the results expressed in log? Check.
AU: We checked all the log values. See in the table 6.
Line 268-270: The coefficient of correlations between bulk tank SCC and milk hygienic quality were as follow: TBC (r = 0.310), CP (r = 0.305) CC (r = 0.232) and CIP (r = 0.215). I suggest the Authors enter the p-value following Pearson's coefficient.
AU: We modified the sentence as suggested. See lines 317 to 319.
Line 373-381: Conclusions must be clearly rewritten.
AU: We rewrote the conclusion to better clarify. See line 424 to 430.

Reviewer 2 Report
Rewiew Animals-2514362
In this paper entitled “Association Between Socioeconomic Profile, Attitudes and Knowledge of Dairy Farmers About Somatic Cell Count and Milk Quality”, the Authors, investigate the association between the bulk tank somatic cells count and the knowledge of dairy farmers about milk quality, mastitis control, and their socioeconomic characteristics, by administering specific questionnaires to farmers. Furthermore, this study estimated the association between bulk tank somatic cells count and the hygienic quality of bulk milk on dairy farms. The socio-economic profile, attitudes and knowledge of the managers of 63 dairy farms, were analysed. The Authors conclude that the economic impact of milk of mastitis, milking procedures, methods for subclinical mastitis detection, and awareness about legal requirements of somatic cell count and total bacterial count were associated with somatic cell count in bulk tank milk of dairy herds.
This paper, in line with the subject matter of the journal, is well articulated in its treatment of the topics, with interest for the scientific community, it needs some additions that I report below line-by-line:
Line 93-95: table 1. Check the measurement intervals in table 1, e.g. 1-50, 50-100, 100-200 ….
Line 99-100: (August and 99 September 2011). Is this date correct?
Line 113-114: "Somatic cell counts were analysed by electronic counting with flow cytometry (Fossomatic FC, Foss Electric A/S Hillerod, Denmark). Add further details, e.g. internal standards, measuring range, ….
Line 169: Check “eij =randon error”.
Line 174: The results are expressed as ...; specify.
Line 194: I suggest the Authors include a short comment on the use of the CMT in the milking parlour (see also line 309).
Line 209: Check “Monthly average income (R$)”.
Line 217: Check “managers (76,2%).”
Line 219: Check “managers (6,3%).”
Line 225-226: “managers (3,2%) associated mastitis with reduced feed intake, 6 (9,5%) associated mastitis 225 with other issues, and 5 (7,9%) were unsure”. Check the %.
Line 263-265: Table 6. Total bacterial count (TBC), preliminary incubation count (PIC), pasteurized count (PC), total coliform count (CC) and somatic cells count (SCC), grouped by somatic cells count. Are the results expressed in log? Check.
Line 268-270: The coefficient of correlations between bulk tank SCC and milk hygienic quality were as follow: TBC (r = 0.310), CP (r = 0.305) CC (r = 0.232) and CIP (r = 0.215). I suggest the Authors enter the p-value following Pearson's coefficient.
Line 373-381: Conclusions must be clearly rewritten.
The paper requires moderate editing of the English language.
Author Response

(The authors gave the same response as above.)
